# Identification of the Transcriptional Regulatory Role of RUNX2 by Network Analysis in Lung Cancer Cells

**DOI:** 10.3390/biomedicines10123122

**Published:** 2022-12-03

**Authors:** Beatriz Andrea Otálora-Otálora, Cristian González Prieto, Lucia Guerrero, Camila Bernal-Forigua, Martin Montecino, Alejandra Cañas, Liliana López-Kleine, Adriana Rojas

**Affiliations:** 1Grupo de Investigación INPAC, Unidad de Investigación, Fundación Universitaria Sanitas, Bogotá 110131, Colombia; 2Facultad de Medicina, Universidad Nacional de Colombia, Bogotá 11001, Colombia; 3Departamento de Estadística, Universidad Nacional de Colombia, Bogotá 11001, Colombia; 4Instituto de Genética Humana, Facultad de Medicina, Pontificia Universidad Javeriana, Bogotá 110211, Colombia; 5Institute of Biomedical Sciences, Facultad de Medicina y Facultad de Ciencias de la Vida, Universidad Andres Bello, Santiago 8370134, Chile; 6Departamento de Medicina Interna, Facultad de Medicina, Pontificia Universidad Javeriana, Bogotá 110211, Colombia; 7Unidad de Neumología, Hospital Universitario San Ignacio, Bogotá 110211, Colombia

**Keywords:** lung cancer (LC), differentially expressed genes (DEGs), transcription factors (TFs), coexpression networks, gene regulatory network (GRN), diagnostic and prognostic biomarkers

## Abstract

The use of a new bioinformatics pipeline allowed the identification of deregulated transcription factors (TFs) coexpressed in lung cancer that could become biomarkers of tumor establishment and progression. A gene regulatory network (GRN) of lung cancer was created with the normalized gene expression levels of differentially expressed genes (DEGs) from the microarray dataset GSE19804. Moreover, coregulatory and transcriptional regulatory network (TRN) analyses were performed for the main regulators identified in the GRN analysis. The gene targets and binding motifs of all potentially implicated regulators were identified in the TRN and with multiple alignments of the TFs’ target gene sequences. Six transcription factors (E2F3, FHL2, ETS1, KAT6B, TWIST1, and RUNX2) were identified in the GRN as essential regulators of gene expression in non-small-cell lung cancer (NSCLC) and related to the lung tumoral process. Our findings indicate that RUNX2 could be an important regulator of the lung cancer GRN through the formation of coregulatory complexes with other TFs related to the establishment and progression of lung cancer. Therefore, RUNX2 could become an essential biomarker for developing diagnostic tools and specific treatments against tumoral diseases in the lung after the experimental validation of its regulatory function.

## 1. Introduction

Lung cancer is a disease characterized by an increase in the growth of pulmonary cells and metastasis to other tissues [1]. Non-small-cell lung cancer (NSCLC) is the histological type that represents 85% of cases of lung cancer [2]. Lung cancer causes 18.4% of diseases related to tumoral pathologies. It is the most frequently diagnosed type of cancer that causes death in men and the second most frequently diagnosed in women [3]. Identifying important and specific biomarkers for developing early diagnostic techniques and treatments against the disease is the key to reducing the high mortality rates associated with lung cancer. In NSCLC patients, a number of predictive biomarkers (BRCA1, TP53, and KRAS) [4] have been identified as possible genetic risk factors that increase their susceptibility to developing lung cancer, which might be necessary for the development of treatments against the disease [5,6]. However, there are only a limited number of NSCLC patients with specific gene variations, and the mortality rates of this cancer have, therefore, not been reduced significantly [7].

The transcriptome reflects the genetics, epigenetics, and microenvironment of the tumor tissue and largely determines the phenotype of tumor cells [8]. We have developed a new bioinformatics pipeline based on the creation of coexpression networks [9] to analyze the complexity of lung cancer by examining a large number of deregulated genes (DEGs) identified in transcriptomic studies [10], investigating the signaling pathways related to oncogenic gene networks, and identifying common connectivity patterns (CCPs) by comparing coexpression networks, which may help to identify promising targets for the pharmaceutical treatment of lung cancer [11]. Our bioinformatics pipeline identified the most important DEGs and transcription factors (TFs) as key biomarkers of lung cancer due to their association with biological processes and signaling pathways related to the acquisition of the hallmarks of cancer during tumor formation in pulmonary tissue [12].

The construction of gene regulatory networks (GRNs) with data from large-scale gene expression studies is considered an important method for understanding a biological system from a global perspective, where associations between vertices and nodes represent biological regulatory interactions between deregulated genes encoding TFs in lung cancer cells and the target genes of TFs [13,14]. The analysis of GRNs based on microarray datasets has demonstrated the ability to identify unique and common biomarkers among different types of cancer, such as the overexpression of RAD51-associated protein 1 (RAD51-AP1) in ovarian and lung cancer in association with the poor prognosis of patients and with increased cell proliferation in cell lines of these two types of cancer, suggesting that RAD51-AP1 is a potential biomarker that can be applied in clinical practice [15]. The use of gene expression studies for the realization of GRNs has enabled differential expression profiles to be utilized for the identification of molecules associated with both dysplasia and MAPK that induce adenocarcinomas, leading to a better classification of dysplasia, an improved understanding of the biological processes involved in the progression of lung cancer, and the validation of targets for the development of treatments [16].

Transcriptomic analysis for the identification of transcription factors in lung cancer was extended beyond coexpression networks [12] to GRNs because they can generate direct information regarding the regulatory interactions of transcription factors and their potential targets [17]. Moreover, a coregulatory network analysis enables the identification of TFs that actively participate in the process reflected in the data, thereby constructing a potential interaction framework for the effects of several regulators as cooperative or coregulatory complexes on the expression of their target genes [18].

This manuscript identifies six relevant transcriptional regulators (E2F3, FHL2, ETS1, KAT6B, TWIST1, and RUNX2) in a GRN as important TFs for NSCLC. On the other hand, the CoRegNet library enabled us to analyze the importance of transcriptional regulators in lung cancer, leading to the identification of RUNX2 as an important transcriptional regulator in the process. Further analysis enabled us to identify many targets and a potential regulatory binding motif. The RUNX2 gene encodes a transcription factor that is the master regulator of the osteogenic differentiation of mesenchymal cells, and its active participation in cancer has been previously described. For lung cancer, RUNX2 is involved in the epithelial–mesenchymal transition process through the genetic control of vimentin, TWIST, and SNAIL [19]. Our results help to characterize RUNX2, indicating that it could be a master regulator and direct regulator of many genes and that other TFs are involved in the regulation of gene expression in lung cancer.

## 2. Materials and Methods

### 2.1. Gene Regulatory Network Analysis (GRN)

The GSE19804 gene expression dataset was utilized for this study. Although smoking is a major risk factor for lung cancer, only 7% of female lung cancer patients in Taiwan have a history of cigarette smoking, which is considerably lower than that reported for Caucasian females. This report is a comprehensive analysis of the molecular signature of nonsmoking female lung cancer patients in Taiwan [20]. This dataset includes different stages (I, II, III, and IV) and subtypes, including adenocarcinoma, bronchioloalveolar carcinoma, and squamous carcinoma. The bioinformatics pipeline used in this work consists of four main steps and is presented in Figure 1. Leal and López-Kleine (2018) [21] proposed a novel method to reconstruct a GRN using microarray data or RNA-Seq count data based on similarity measurements and organized the method into the following steps.

The first step is to select a measurement to estimate the similarity between each pair of genes or “gene expression profiles” from the expression matrix E. Spearman correlation and Kendall correlation can be used on data with unknown distributions. Mutual information is a nonlinear dependence measure and can also be used [22]. The result is a similarity matrix S that shows the intensity of the similarity and the sign of the linear correlation, which can provide information regarding the direction of the process. This proposal is based on a previous methodology developed and indicated to work well for similarities obtained on continuous microarray data and for the construction of gene coexpression networks (non-signed edges, only indicating a coordinated activity of genes) [23]. The steps for the threshold selection were the following:A list of threshold values to be evaluated is created.

b.For each threshold, a different adjacency matrix A is constructed for the similarity matrix *S* using Equation (1).

Equation (1): Equation for matrix A construction. $s_{ij}$ is the similarity measurement selected, $a_{ij}\;$is the value in the adjacency matrix in row i and column j, and $\tau^{*}$ is the specific threshold proposed.
(1)$$\tau \left(s_{ij}\right)=\{\begin{array}{c} a_{ij}=1,\;\mathrm{if}\;\left|s_{ij}\right|\geq \tau^{*}\;\mathrm{and}\;s_{ij}>0\\ a_{ji}=0,\;\mathrm{if}\;\left|s_{ij}\right|\geq \tau^{*}\;\mathrm{and}\;s_{ij}>0\\ a_{ij}=0,\;\mathrm{if}\;\left|s_{ij}\right|\leq \tau^{*}\;\mathrm{and}\;s_{ij}<0\\ a_{ji}=1,\;\mathrm{if}\;\left|s_{ij}\right|\leq \tau^{*}\;\mathrm{and}\;s_{ij}<0 \end{array}$$

c.For each adjacency matrix A, the node clustering coefficient is calculated [24]. The expected clustering coefficient for a random graph with the same characteristics is also calculated.d.The absolute values of the differences between the (under randomness) expected and observed cluster coefficients are calculated and plotted. The threshold is chosen at the point where the difference between the two is the greatest.e.Finally, the adjacency matrix for the selected threshold is constructed.

The adjacency matrix is nonsymmetric and determines a directed network. The method was implemented in R [25] and was used to generate a GRN of non-small-cell lung cancer (NSCLC). The standardized gene expression levels of the DEGs from the microarray dataset GSE19804 were used to construct a similarity matrix using Spearman’s correlation. The similarity threshold was identified based on the methodology established by Leal et al. (2014), and an adjacency matrix was constructed from the similarity matrix to create a directed network [26]. The clustering coefficient of the array was calculated using a random adjacency matrix to calculate the expected clustering coefficient. The difference between the clustering coefficients for each threshold was plotted, and the threshold with the highest absolute value was subsequently taken in the random network and the expression data network, thereby defining the adjacency matrix for the construction of the regulatory network [27]. For all genes, the degree was established (this indicates the number of connections), and they were sorted from the highest to the lowest degree. This measure of centrality is important for the genes in the network. 

### 2.2. Gene Coregulatory Network Analysis

The same database (GSE19804) was also used to construct a coregulatory network with the R library “CoRegNet” [8] for TFs identified in the GRN [12]. The coregulatory analysis enabled us to identify which of the TFs could be associated with NSCLC lung cancer as coregulators. 

### 2.3. Identification of Potential Gene Targets of RUNX2

The reported target genes of the most important TF based on all of the above-gathered information (deregulation in other lung diseases, coregulation role, and high degree in coexpression network) were searched for in five databases: Jolma (http://fiserlab.org/tf2dna_db//search_genes.html) (accessed on 30 June 2019), Jaspar (http://jaspar.genereg.net) (accessed on 30 June 2019), Hocomoco (http://autosome.ru/HOCOMOCO/, http://cbrc.kaust.edu.sa/hocomoco/) (accessed on 30 June 2019), Trust (https://www.grnpedia.org/trrust/) (accessed on 30 June 2019), and HumanNet (http://www.functionalnet.org/humannet/cgi-perl/HumanNet.v1_nga_form.cgi) (accessed on 30 June 2019), which use different methodologies to identify the target genes of a TF based on significant *p*-values and scores that validate their association. The list of all possible genes regulated by RUNX2 according to the databases was searched against the list of differentially expressed genes in the microarray datasets in which RUNX2 is positively regulated (GSE19803, GSE108055, and E-MTAB-5231) [12] to identify potential functional targets of RUNX2 as those that might be deregulated by the presence of the TF. The functional targets were validated with the datasets of lung cancer in which RUNX2 is not differentially expressed (GSE3268, E-MTAB-3950, and GSE10072) [12] to identify the direct functional targets of the transcription factor RUNX2 as those that might show a change in their regulation in the absence of TF expression in the affected lung tissue. The database (GSE19804) was also used to construct a transcriptional regulatory network of the main regulators identified in the RTN library [28] in order to analyze their cooperative function in regulating the same target genes. DAVID Functional Annotation and Enrichment Analysis [29] and the PANTHER database [30] were used on all possible target genes of RUNX, including those identified by the transcriptional regulatory network, to identify those with evidence related to oncogenic signaling pathways. 

### 2.4. Identification of Potential Binding Motifs of RUNX2

The identification of binding motifs was performed through multiple alignments of the sequences of the TF target genes identified in the previous steps. To this end, 1000 nucleotide sequences prior to the start of every target gene were obtained from the NCBI database and introduced in the software MEME (Multiple EM for Motif Elicitation) (http://meme-suite.org/tools/meme) (accessed on 11 July 2019). Based on preliminary searches, the parameters were set to the search of a maximum of two sequences and only one motif in each sequence with a length between 6 and 20 nucleotides; other parameters were left as default values. To identify these motifs de novo, the program uses position-specific weight matrices that determine the probability of every letter of a possible sequence. The results were subsequently used as input for Tomtom (http://meme-suite.org/tools/tomtom) (accessed on 11 July 2019), to compare these motifs with those already reported in the Jaspar Core database (http://jaspar.genereg.net) (accessed on 11 July 2019), specifically in the human genome. Parameters in Tomtom were kept as default values. For this comparison, only significant motifs were considered, which are those determined by a distance value between the query and the target sequence according to an expected probability distribution function.

## 3. Results

### 3.1. Six Transcription Factors (E2F3, FHL2, ETS1, KAT6B, TWIST1, and RUNX2) Are Essential Regulators of Gene Expression in NSCLC

In previous studies conducted by our research group [12], ten microarray datasets (six of lung cancer and four of lung diseases) were selected to identify hub differentially expressed genes (DEGs) shared by lung pathologies and lung cancer. For the identification of the most important transcription factors in NSCLC lung cancer, GSE19804 was selected by applying quality criteria. Another consideration that affected the selection of this single dataset was the fact that this analysis considers subtype differences (e.g., adenocarcinoma, bronchioloalveolar carcinoma, and squamous carcinoma) and stages I, II, III, and IV [20]. 

The NSCLC lung cancer regulatory network (NSCLC-GRN) has a total of 38 nodes (Figure 1). Ten nodes were determined to be the most highly connected, of which five (COL1A1, SHMT2, ATP2A2, MCM6, and PCNA) were also identified as relevant genes in previously constructed coexpression networks [12]. Our previous coexpression network analysis identified several TFs associated with the most common DEGs based on the joint analysis of lung cancer and other lung disease datasets and some common connection patterns between coexpressed genes. The NSCLC-GRN identified six important transcriptional regulators (E2F3, FHL2, ETS1, KAT6B, TWIST1, and RUNX2) (Figure 1), which were considered for the posterior coregulatory analysis.

### 3.2. RUNX2 Is an Important Regulator and Coregulator in NSCLC

Important regulatory mechanisms for the establishment and progression of lung cancer have been associated with the 26 TFs identified in our previous coexpression network analysis [12,19,31] and with 6 TFs identified in the analysis of NSCLC lung cancer regulatory networks conducted in this study (Figure 1). The CoRegNet library generated a lung cancer coregulatory network with the gene expression levels of the GSE19804 dataset to assess the importance of the six transcriptional regulators found in the NSCLC-GRN in lung cancer establishment and progression and to identify a transcriptional regulator capable of participating as a coregulator of the tumor process via functional association with the most critical TFs identified in the gene coexpression network analysis [12].

The coregulatory network analyzed the cooperative function of the TFs of the NSCLC regulatory network (Figure 2), along with the transcription factors identified in our previous coexpression network studies (Table 1) [12,31]. The first coregulatory network suggests a cooperative function between ETS1 and KAT6B with seventeen transcription factors of our previous coexpression analysis. The second suggests a cooperative function between RUNX2 and three transcription factors (BRCA1, FOXM1, and RUNX1) important for the specific regulation of lung cancer establishment and progression [12,31], along with three transcription factors (E2F3, FHL2, and TWIST1) of the NSCLC-GRN (Figure 1). Two of the six transcriptional regulators (RUNX2 and TWIST1) are deregulated in other lung diseases (LD) (i.e., idiopathic pulmonary fibrosis (IPF) and pulmonary arterial hypertension (PAH)). RUNX2, but not TWIST1, can function as a coregulator of BRCA1, a TF associated with the regulation of common connectivity patterns between LC-PAHs [12]. RUNX2 also acts as a coregulator of FOXM1, the TF that regulates lung cancer CCPs, and RUNX1, a TF that is coexpressed in the networks of unique lung cancer genes that are not deregulated in other lung diseases (LCI) or other types of cancer (LCII) [31]. RUNX1 is a TF identified in LCI and LCII networks, which were selected to focus the analysis on lung cancer [12,31]. Similarly, RUNX2 is a coregulator of three TFs (E2F3, FHL2, and TWIST1) that also appear in the NSCLC lung cancer regulatory network (Figure 2).

### 3.3. Potential Target Genes of RUNX2

One hundred and fifty-six putative genes regulated by RUNX2 were identified in the databases, of which seventy-four might be functional targets, and twenty-seven might be direct targets of RUNX2 (Table 2). DAVID Functional Annotation and Enrichment Analysis showed that fifty RUNX2 targets are statistically significantly associated with cancer (pc value: 1.3 × 10^−4^), of which nineteen are specifically associated with lung cancer (pc value: 1.5 × 10^−4^), and some are associated with signaling pathways such as TGF-beta (pc value: 1.3 × 10−^10^), Hippo (pc value: 1.3 × 10^−6^), Wnt (pc value: 3.0 × 10^−3^), Notch (pc value: 2.9 × 10^3^), and the regulation of pluripotency in stem cells (pc value: 3.7 × 10^−6^). 

Four hundred and thirty-two genes were identified as potential RUNX2 targets in the TRN analysis, of which one hundred and one are significantly related to cancer (pc value: 4,1E−2), twenty-two have been specifically associated with lung cancer, and nineteen (CD19, COL1A1, COL1A2, COL6A3, COMP, EFNA4, FGFR4, ITGA11, ITGAV, LAMA1, LAMC3, LPAR2, LPAR5, MAGI1, PDGFC, PPP2R5A, SPP1, THBS2, and TNC) are related to the PI3K-Akt signaling pathway. According to Panther, there are seventeen RUNX2 target genes (ACTN1, COL10A1, COL11A1, COL1A1, COL1A2, COL3A1, COL5A1, COL5A2, COL6A3, COL8A2, ITGA11, ITGAV, LAMA1, LAMC3, RAP2B, RRAS, and SLK) related to the integrin signaling pathway (P00034) and nine (LEF1, LRP5, MMP7, PCDH9, PPP2R5A, PRKCE, SDK1, TCF3, and TLE1) related to the Wnt signaling pathway (P00057). 

Furthermore, the transcriptional regulatory network of lung cancer built with the RTN library (Figure 3) allowed the identification of the total number of target genes of the transcriptional regulator RUNX2 and its six coregulators, E2F3, FHL2, TWIST1, BRCA1, FOXM1, and RUNX1, divided into the number of targets that each TF regulates positively and negatively (Table 3). The TFs regulate a larger number of genes positively than negatively and regulate a substantial number of the same genes. Two of the targets of RUNX2 in the RTN network are FOXF1 and E2F3. FOXF1 is a negatively regulated TF in eight datasets of lung cancer and in the PAH dataset [12], which appear to be coexpressed in the LCII network (Table 1) [31]; meanwhile, E2F3 is overregulated in seven datasets of lung cancer [12] and is one of the six important transcriptional regulators identified with RUNX2 in the NSCLC-GRN (Figure 1).

### 3.4. Potential Binding Motifs of RUNX2

The identification of the binding motifs was performed through multiple alignments of the sequences of the RUNX2 target genes identified in the previous steps. Two motifs were identified as possible common binding sites of RUNX2 among its functional targets (Figure 1). Both motifs have high significance and were validated and compared to previously reported motifs. After the comparison was performed, we observed that both motifs resemble the forward and reverse sequences, respectively, of the binding site of a C2H2 zinc-finger protein, ZNF263, which is reported to be involved in different types of cancer, although there is no record to date of its activity in lung cancer.

## 4. Discussion

The search of the gene regulatory network of lung cancer started with a joint transcriptomic analysis of lung cancer and other lung diseases to identify common DEGs between lung cancer and other lung diseases, along with unique DEGs in lung cancer that are only deregulated in tumor cells [12]. Next, we performed another transcriptomic analysis comparing lung cancer with other types of cancer to identify common DEGs between lung cancer and other types of cancer, along with unique DEGs in lung cancer that are only deregulated in lung tumor cells [31]. Both studies identified coexpression networks of unique lung cancer genes that are not deregulated in other lung diseases (LCI) or other types of cancer (LCII), among which there are transcription factors (Figure 2) that share functions related to the acquisition of the hallmarks of cancer [12,31]. Likewise, the most important transcription factors capable of regulating each of the coexpression networks were identified (Table 1). There are two previous studies in lung adenocarcinoma (LUAD) and lung squamous cell carcinoma tissues (LUSC), which analyzed gene coexpression in microarray datasets. An analysis of RUNX2 mRNA expression levels in LUAD with immunohistochemistry, high-throughput RNA sequencing, and gene microarrays identified DEGs in LUAD, potential target genes of RUNX2, and its coexpressed genes promoting cell proliferation and drug resistance in LUAD by modulating the cell cycle and MAPK signaling pathways [32]. An integrated expression analysis in LUSC has been previously conducted with RNA-Seq and microarray data from multiple platforms, generating a coexpression network of RUNX2 and 45 deregulated genes, which were significantly clustered in pathways, including the PI3K-Akt signaling pathway, playing a key role in the clinical progression of LUSC [33]. 

The González and López-Kleine (2018) method [27], developed by our research group, was used to construct the NSCLC-GRN in order to identify important transcriptional hubs and regulatory genes associated with important mechanisms of gene expression control in lung cancer. The gene regulatory network (NSCLC-GRN) identified six regulators (Figure 1) for which there is evidence of their ability to form coregulatory complexes with each other and other transcription factors important for specific lung cancer establishment and progression (Figure 2), and probably with other cofactors [34,35,36], to fulfill their regulatory function as a repressor or activator of gene expression in lung cancer. 

In our two transcriptomic studies, we addressed tumor heterogeneity at different levels to understand the sophisticated gene interactions and regulatory coordination that ultimately led to the formation of a specific tumor phenotype. Our main goal was to identify a characteristic meta-signature of lung cancer as a general disease. We first identified DEGs in each type and subtype of lung cancer in different datasets to find the genes deregulated in most of them, and then we filtered out those that are regulated in other types of lung diseases and other types of cancer to form coexpression networks of genes specifically related to lung cancer. We also addressed this issue by comparing the coexpression networks of lung cancer, other lung diseases, and other types of cancer to identify common connectivity patterns (CCPs) in Coexnet or groups of genes that have the same patterns of expression and molecular interactions in lung cancer and different types of lung diseases, suggesting that they could be related to a pre-tumor state; in different types of cancer, suggesting that they could be related to a general tumor state; and in different types and subtypes of lung cancer, suggesting that they could be related to a specific lung tumor state and ultimately related to a cooperative and coordinated biological function during the acquisition of the hallmarks of cancer. 

There are eleven publications assessing the transcription regulatory network of lung cancer with different microarray and RNA-Seq studies, performing direct and very different bioinformatics analyses on datasets created or selected for publication [37,38,39,40,41,42,43,44,45,46,47]. None attempted to perform a general analysis of lung cancer, and most conducted the analysis with a reduced number of datasets and for each subtype of lung cancer independently. None of them tried to select deregulated genes unique to lung cancer that are not deregulated in other lung diseases or other types of cancer, and none performed a joint coregulatory analysis to study the cooperative and coordinated regulatory functions of transcription factors. Four studies were conducted with cell lines [37,38,39,40], five were conducted with one, two, or six microarray studies [41,42,43,44,45], and two were conducted with TCGA RNA-Seq studies [46,47]. However, some of the transcription factors identified using our bioinformatics pipelines are also important in some of the other studies (FOXM1, RUNX1, HES1, TAL1, FOXF1, E2F1, MYBL2, IRF1, STAT6, NR4A2, EBF1, ZEB1, MEF2A, and YY1) (Table 4). Therefore, regardless of the cell types used, the gene expression study used, and the bioinformatics methodology used, there is a group of regulatory genes that consistently occur in a statistically significant manner in lung cancer.

CoRegNet allowed us to analyze the importance of transcriptional factors as coregulators, leading to an integrated analysis of coexpression and regulatory networks (Table 1), which highlight RUNX2 as a possibly important transcriptional regulator in NSCLC lung cancer. The transcriptomic analysis previously identified RUNX2 as a deregulated gene in the PAH dataset, and RUNX2 also appears in the LC-PAH CCP, suggesting that it is also a critical TF for other lung diseases. TWIST1 was also deregulated in IPF and PAH; however, it did not appear in any LC-LD CCPs; therefore, the evidence of its association with lung cancer is not as strong as it is for RUNX2 [12].

RUNX2 is a coregulator of BRCA1 (Figure 2), the TF that regulates the LC-PAH CCP, strengthening the evidence that RUNX2 is associated with pulmonary arterial hypertension and lung cancer [12]. RUNX2 is also a coregulator of FOXM1 (Figure 2), the TF that regulates all LC CCPs, thereby suggesting that RUNX2 is associated with lung cancer development, as it participates in the regulation of important genes coexpressed in several lung cancer networks [31]. Similarly, RUNX2 is also a coregulator of RUNX1 (Figure 2), a TF of the same family, and is coexpressed in coexpression networks of unique lung cancer DEGs that are not dysregulated in other lung diseases (LCI) [12] or in other types of cancer (LCII) [31], with which it shares binding motifs with its target genes (Figure 3). Moreover, RUNX2 is a coregulator of three of the transcriptional regulators identified in the NSCLC-GRN (i.e., E2F3, FHL2, and TWIST1) (Figure 2), suggesting a cooperative function of these TFs in lung cancer, which has not been studied experimentally in lung cancer.

RUNX2 is considered the master regulator of the formation of the osteoblastic lineage, directly or indirectly controlling the expression of several key genes (collagen 1, osteocalcin, osteopontin, alkaline phosphatase, and bone sialoprotein) for the early and late differentiation of osteoblasts [48]. Similarly, BRCA1 has proven to be indispensable for osteoblastic proliferation [49]. The association of RUNX2 and BRCA1 with the osteoblastic lineage may suggest that its dysregulation in lung cancer may occur in the formation of coregulatory complexes, thereby affecting the function of multiple essential genes in lung pathologies. BRCA1 controls the expression of miR-155 at the epigenetic level, which, in turn, directly regulates FOXO3a and RUNX2, affecting the metastatic potential of breast tumor cells [50]. In breast cancer, RUNX2 and FOXM1 induce EMT and are regulated by sumoylation [51], and they share DNA binding sites in ERα; this association leads to the activation of genes involved in proliferation and metastasis and the development of endocrine resistance during tamoxifen treatment [52].

The coregulatory function between RUNX2 and RUNX1 is associated with the process of dynamic transfer to the same positionally stabilized nuclear foci, dependent on the c-terminus, which is essential for the assembly of macromolecular complexes of coregulatory proteins at sites associated with the nuclear matrix; thus, RUNX2 and RUNX1 regulate gene transcription in specific tissues [53]. The coregulatory function between RUNX2 and FHL2 has not been studied, and the only evidence of their association is related to processes driving the differentiation of mesenchymal cells into osteoblasts [54]. The coregulatory function between RUNX2 and E2F3, as well as RUNX2 and TWIST1, has not been studied either, as there are no studies suggesting their association. However, it has been observed that TWIST1 can inhibit osteoblastic differentiation by negatively regulating RUNX2 [55]. In our laboratory, RUNX2 has been related to the positive regulation of gene expression associated with EMT, such as vimentin and SNAIL1, thereby increasing the migratory capacity of lung adenocarcinoma cells [19]. RUNX2 expression has been correlated with tumor size, tumor stage, lymph node metastasis, and shorter postoperative survival time, suggesting that it might become a novel prognostic marker in NSCLC [56].

The transcriptional regulatory function of RUNX2 in lung cancer could be associated with the ability to participate as a coregulator of important TFs and cofactors, which has also been identified in our previous analysis of coexpression networks in lung cancer [12,31], and the epigenetic regulation of its expression has also been studied [19]. The regulatory network showed an essential association with signaling pathways related to the acquisition of cancer characteristics and the regulatory mechanisms involved in establishment and progression, such as PI3K-AKT, WNT, PLK1, cGMP-PKG, and p53 (Figure 4). Five (COMP, FGFR4, ITGA11, LPAR2, and PPP2R5A) of the nineteen RUNX2 target genes related to the PI3K-Akt signaling pathway are coexpressed in the CCP formed between lung cancer and PAH [12], and four (COL3A1, ITGA11, RAP2B, and PPP2R5A) of the nine RUNX2 target genes related to Wnt signaling pathway are coexpressed in the CCP formed between lung cancer and PAH, suggesting the importance of the RUNX2 regulatory function in these two signaling pathways in lung cancer establishment.

The databases identified 156 possible RUNX2 target genes, some of which are significantly associated with transcriptional dysregulation in cancer, proteoglycans in cancer, angiogenesis, and tumor cell growth, as well as the acquisition of cancer characteristics [63], suggesting their importance in the establishment and progression of lung tumors. Some RUNX2 target genes are associated with two signaling pathways. The Hippo signaling pathway inhibits the YAP and TAZ transcriptional coactivators by phosphorylation, while the TGF-beta signaling pathway positively regulates TAZ, which interacts with RUNX2 by coactivating it to promote cell differentiation in mesenchymal stem cells [64] and thus induces a tumor phenotype [65], while YAP can act as an inhibitor of RUNX2 [66]. NOTCH1 binds to RUNX2 to directly inhibit it and upregulates CDK4, which ubiquitinates and degrades RUNX2 [67], thereby controlling the lineage specification [68]. The participation of RUNX2 in signaling pathways associated with stem cell pluripotency suggests the participation of this TF in the differentiation of tumor stem cells (CSCs) towards tumor cells (CCs) in specific tissues.

The RUNX family of transcription factors forms a heterodimeric complex with CBFβ, which increases its DNA binding affinity by exposing its DNA binding domain [69]. RUNX2 forms a complex with p53 and histone deacetylase 6 (HDAC6) in the cell nucleus following DNA damage, inhibiting transcriptional and pro-apoptotic activity [70]. Zfp521 is a binding partner of Runx2 during the early stages of osteoblast differentiation [71]. The MEF and Runx2 proteins form a complex, which interferes with Runx2 binding to the cis-acting element OSE2, promoting cell proliferation and inhibiting osteogenic differentiation [72]. ATF4 and ZFP521 act as cofactors of Runx2 during bone development, regulating specific transcriptional cascades [71]. 

ETS1 is a target of RUNX2 in both types of lung cancer (Table 3) [12]. ETS1 is not dysregulated in other types of cancer [31]; it is only dysregulated in lung cancer, suggesting that it is a crucial gene for lung tumor progression. Likewise, ETS1 is one of the transcriptional regulators identified in the regulatory network in NSCLC lung cancer (Figure 1). The association of ETS1 as a direct target of RUNX2 in lung cancer strengthens the evidence that RUNX2 is a critical transcriptional regulator since the coregulation analysis showed that ETS1 is a coregulator of a significant number of TFs identified in the networks of coexpression (Table 1) (Figure 2), among which are NR4A2 and ZEB1, two important transcription factors for the establishment and progression of the tumor process in the lung that appeared in several networks coexpressed with unique DEG hits for lung cancer [12]. Therefore, RUNX2 indirectly regulates the TF regulatory network identified by coexpression analysis, especially NR4A2 and ZEB1, through the direct regulation of ETS1. MiR-9 suppresses the expression of cyclin D1 and Ets1 by binding to their 3′-UTRs; it has been seen that it leads to the inhibition of proliferation, invasion, and metastasis processes in tumor cells [73]. In lung cancer, RUNX2 directly inhibits ETS1 expression, which is likely to promote angiogenesis, proliferation, invasion, and metastasis. In NSCLC, the microRNA-130a-5p/RUNX2/STK32A network modulates tumor proliferation, metastasis, and invasion. MiR-130a-5p must be downregulated in order to maintain RUNX2 overexpression and interaction with STK32A [74]. In thyroid cancer cells, HDAC6 potentiates RUNX2 transcription, stabilizing the assembly of the transcriptional complex on the RUNX2 P2 promoter [75]. In breast cancer, ABL phosphorylates RUNX2 through direct binding, activating its expression through its SH2 domain in a kinase-activity-dependent manner, and the activation the bone morphogenetic protein (BMP)-SMAD pathway, promoting tumor cell invasion [76]. MiR-196b must be significantly decreased in lung cancer to maintain tumor cell viability, migration, and invasion and EMT induction by TGF-β, PI3K/AKT/GSK3β, Smad, and JNK pathways, as Runx2 is a putative target of miR-196b [77]. The WWOX mRNA level is downregulated in tumor tissues by hypermethylation deletion and destabilizing mutations, or it would inhibit RUNX2, reducing the invasiveness of lung cancer cells [78].

RUNX2 is known for its importance in the progression of cells through the osteoblastic lineage from pluripotent mesenchymal cells to mature osteocytes, and this protein is regulated by multiple physiological signals, including transforming growth factor TGF-beta, vitamin D, glucocorticoids, and bone morphogenetic protein (BMP) [79,80]. BMP/TGF-beta signaling is vital in the maintenance of the typical pulmonary arteriolar structure [81,82] and in the abnormal function of BMPR2, one of the “guardians” of the homeostasis of the pulmonary vessels during process repair. This signaling controls apoptosis and cell proliferation in pulmonary vascular smooth muscle and endothelial cells [83].

The transcription factors of the RUNX family have been shown to be essential regulators of developmental cell fate and to have the ability to act as both dominant oncogenes and tumor suppressors in cancer progression [84]. RUNX proteins have some common characteristics in their transactivation/inhibition domains and in some specific conserved motifs, such as the nuclear-matrix binding site and the VWRPY motif that interacts with corepressors [85]. RUNX2, as a member of the RUNX family, could act as an activator or repressor of the expression of a particular target gene, depending on the coactivators or corepressors with which it interacts, since RUNX proteins can bind and recruit a broad group of coactivators or corepressors and regulate the promoters of their target genes [86]. RUNX2 complexes might be able to regulate the expression of multiple genes by binding to their promoters or enhancers [87]. The regulation of RUNX2 complexes is lineage- and stage-specific and affects crucial decisions, such as between stopping the cell cycle and continuing proliferation and between differentiation and self-renewal [12,88]. 

We have used a microarray study due to its large number of samples, cases, and controls compared to other datasets. However, microarray studies have a specific number of probes; therefore, not all differentially expressed genes in the samples may have been detected. In addition, we carried out an analysis at the transcriptomic level, where only the transcription factors regulated at the transcriptional level are considered, losing, for example, those that are differentially regulated at the post-translational level. However, we considered both positively and negatively regulated transcription factors throughout the bioinformatics methodology, as they may have an oncogenic or tumor suppressor function, and we supported the analyses of the evidence collected in our previous coexpression studies and the analysis of transcriptional coregulation. Finally, it is important to continue with the experimental validation of the transcription factor network found with the bioinformatics pipeline. There are other studies that included a network-based approach and integrated driver somatic RUNX2 mutations with LINC01614, a known long non-coding RNA (lncRNA) with a putative regulatory role in lung cancer [47].

## 5. Conclusions

In this study, we addressed the complexity of the regulatory mechanisms at the transcriptional level, which are involved in the tumor process in the lung, identifying a network of transcription factors (TFs) that are associated with the acquisition of the hallmarks of lung cancer. The regulatory network analysis identified six transcriptional (E2F3, FHL2, ETS1, KAT6B, TWIST1, and RUNX2) regulators in an NSCLC lung cancer population of female non-smokers, among which RUNX2 seems to be a very important regulator due to its ability to act as a coregulator of important TFs (E2F3, FHL2, TWIST1, BRCA1, FOXM1, and RUNX1) identified in our NSCLC-GRN and previous coexpression networks made with unique deregulated and coexpressed genes in lung cancer, as well as common connectivity patterns between lung cancer, other lung diseases, and other types of cancer, which assess the heterogeneity of lung cancer as a general disease. RUNX2, with its potential target genes, participates as a terminal regulator of lung cancer in the most important signaling pathways (PI3K-AKT, WNT, PLK1, cGMP-PKG, and P53) associated with lung cancer. This result supports previous evidence and helps to further elucidate the role played by this transcriptional regulator in lung cancer as an important coregulator in the tumorigenic process. The analysis of the expression levels of the direct targets of RUNX2 identified ETS1, a transcriptional regulator of lung cancer that appears in the NSCLC regulatory network and is a coregulator of a significant number of TFs identified in the coexpression networks, which are associated with lung cancer. 

The in-silico analysis identifying RUNX2 as an important deregulated gene in NSCLC suggests the possibility of further research aimed at experimental biological validation. The performance of functional assays of loss and gain of function of RUNX2 in NSCLC cells is essential to confirm the conclusions presented in this manuscript.

## Figures and Tables

**Figure 1 biomedicines-10-03122-f001:**
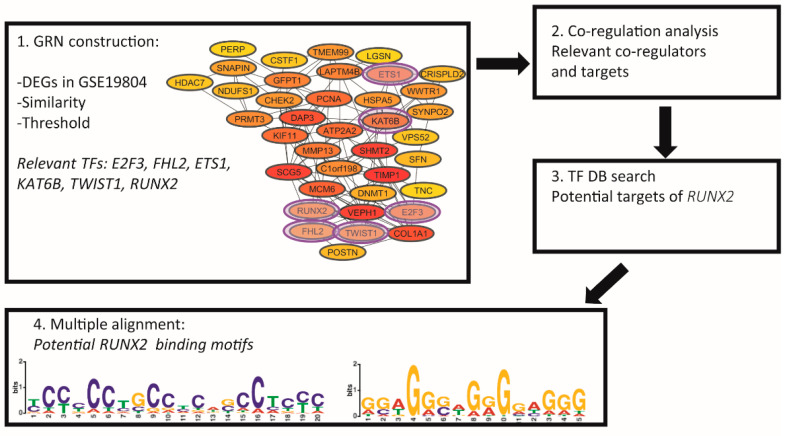
Bioinformatics pipeline used in this work and results in every step: 1. Gene regulatory network (GRN) construction using an in-house methodology. 2. Coregulatory analysis of the main transcription factors (TFs) identified in step 1. 3. Database (DB) search of the reported targets of the most relevant TF (RUNX2) identified in step 2. 4. Identification of potential binding motifs of RUNX2 based on the targets.

**Figure 2 biomedicines-10-03122-f002:**
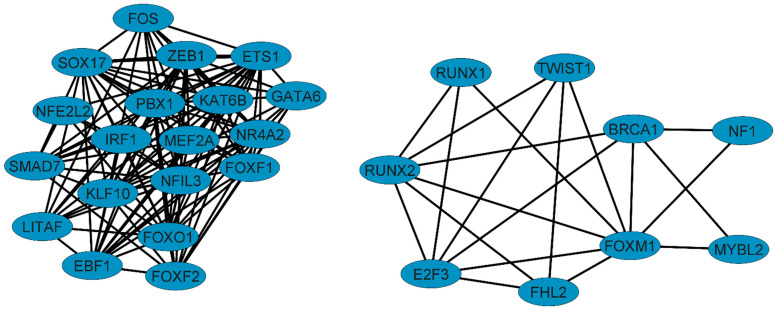
Coregulatory networks of transcription factors related to the establishment and progression of NSCLC lung cancer.

**Figure 3 biomedicines-10-03122-f003:**
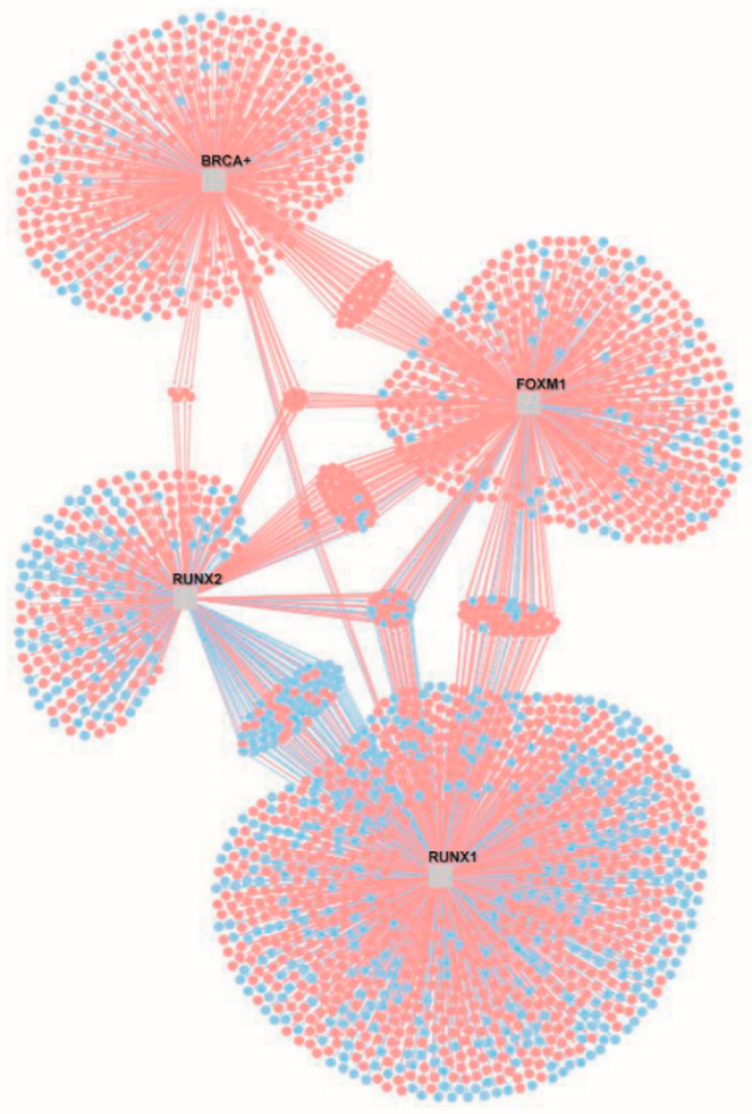
Transcriptional regulatory network of RUNX2 and its coregulators BRCA1, FOXM1, and RUNX1 (gray squares). In blue are the downregulated targets, and in red are the upregulated targets.

**Figure 4 biomedicines-10-03122-f004:**
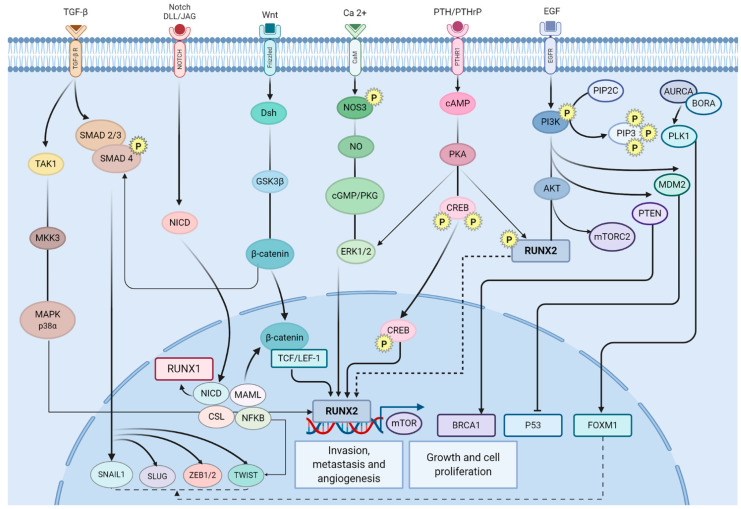
Transcriptional regulators in lung cancer and their association with the acquisition of the hallmarks of cancer and the signaling pathways associated with the establishment and progression of lung cancer. Figure adapted from Figure 1 in Ref. [57], Figure 2 in Ref. [58], Figure 5 in Ref. [59], Figure 3 in Ref. [60], Figure 1 in Ref. [61], and Figures 1 and 3 in Ref. [62].

**Table 1 biomedicines-10-03122-t001:** Main results of our previous bioinformatics analysis [12,31] related to the six regulators in the gene regulatory network of non-small-cell lung cancer (NSCLC-GRN). First, other lung diseases (LD) (IPF (idiopathic pulmonary fibrosis); PAH (pulmonary arterial hypertension)) in which the NSCLC-GRN regulators are deregulated. Then, the transcription factors that regulate common connectivity patterns (CCPs) and the networks of unique lung cancer genes that are not deregulated in other lung diseases (LCI) or other types of cancer (LCII) and the transcription factors in the NSCLC-GRN that can act as coregulators of every NSCLC-GRN regulator.

Transcription Factor	LD	LD CCPs	LC CCPs	LCI	LCII	NSCLC-GRN
E2F3	-	BRCA1	FOXM1	RUNX1	RUNX1	RUNX2, FHL2, TWIST1
FHL2	-	-	FOXM1	-	-	RUNX2, TWIST1
ETS1	-	IRF1, NR4A2, ZEB1		SOX17, FOS, FOXO1, KLF10, SMAD7 LATIF, NR4A2, ZEB1, EBF1	GATA6, FOXF2, FOXF1, NFIL3, NFE2L2, PBX1, MEF2A, EBF1	KAT6B
KAT6B	-	NR4A2, ZEB1		SOX17, FOXO1, KLF10, SMAD7	FOXF1, FOXF2, GATA6, PBX1	EST1
TWIST1	IPF PAH	-	FOXM1	-	-	RUNX2, FHL2, E2F3
RUNX2	PAH	BRCA1	FOXM1	RUNX1	RUNX1	E2F3, FHL2, TWIST1

**Table 2 biomedicines-10-03122-t002:** Potential direct targets of RUNX2 deregulated in lung cancer (LC), non-small-cell lung cancer (NSCLC), and small-cell lung cancer (SCLC) and the signaling pathways associated with its regulatory function.

RUNX2 Targets	Dysregulation	Signaling Pathways
ALYREF	NSCLC	RNA Polymerase II Transcription, TAP/NFX1 pathway
C1orf198	LC	Cell-mediated immune response pathway
DFNB59	SCLC	Afferent auditory pathway
ELANE	NSCLC	Extracellular matrix organization, Innate Immune System
ERP27	SCLC	Photodynamic therapy-induced unfolded protein response
ESR1	SCLC	ERBB4, RNA Polymerase II Transcription, estrogen signaling pathway
ETS1	LC	PDGF, RAS, VEGF, MAPK signaling pathway
HDAC5	SCLC	RNA Polymerase II Transcription, Notch singling pathway
HES1	SCLC	RNA Polymerase II Transcription, angiogenesis, Notch singling pathway
HNRPU	SCLC	mRNA Splicing, lncRNA in canonical Wnt signaling pathway
KIAA1107	SCLC	-
LRP5	LC	Negative regulation of TCF, lncRNA in canonical Wnt, Wnt signaling pathway
MYST4	SCLC	HATs acetylate histones, chromatin organization, P53 signaling pathway
NAGK	SCLC	Synthesis of substrates in N-glycan biosynthesis, metabolism of proteins
NR0B2	SCLC	RNA Polymerase II Transcription, nuclear receptor transcription pathway
PRUNE	SCLC	-
R3HDML	NSCLC	-
REM2	SCLC	-
RNF145	LC	NFκB signaling pathway
SNAPIN	NSCLC	Trans-Golgi Network Vesicle Budding, vesicle-mediated transport
SYNPO2	NSCLC	Nuclear import pathway
TLE1	SCLC	TCF-dependent, Hedgehog, Notch, Wnt signaling pathway
TMEM99	NSCLC	-
VEPH1	LC	TGF-beta, Wnt signaling pathway
YAP1	SCLC	ERBB4, RNA Polymerase II Transcription, transcriptional regulation by RUNX2
ZNF436	LC	RNA Polymerase II Transcription, gene expression (transcription)
ZNF585A	SCLC	RNA Polymerase II Transcription, gene expression (transcription)

**Table 3 biomedicines-10-03122-t003:** According to the RTN library analysis, target genes of the RUNX2 transcriptional regulator and its coregulators, E2F3, FHL2, TWIST1 BRCA1, FOXM1, and RUNX1.

TF	Targets		Total
	Upregulated	Downregulated	
RUNX2	308	124	432
BRCA1	465	65	530
FOXM1	1559	938	2497
RUNX1	1725	912	2637
E2F3	548	287	835
FHL2	362	296	658
TWIST1	256	85	341

**Table 4 biomedicines-10-03122-t004:** Previous gene transcriptional and regulatory network analyses of lung cancer.

Article Title	Lung Cancer	Highlighted Genes
Topological signatures in regulatory network enable phenotypic heterogeneity in small cell lung cancer (2021).	SCLC	TP53, RB1, ASCL1, NEUROD1, YAP1, and POU2F3
A global view of regulatory networks in lung cancer: An approach to understand homogeneity and heterogeneity (2016).	LAD SCC LCC SCLC	HLTF, FOXM1, ARNTL2, LAU, ZNF187, HNRPK, C1orf107, GRLF1, HMGA1, E2F6, IRF1, TFDP1, SUV39H1, RBL1, STAT5A, and HNRPD
A Transcriptional Network Signature Characterizes Lung Cancer Subtypes (2011).	LAC SCC	ABCC3, CLDN3, DPP4, MUC3B, MUC5B, NTRK2, SPINK1, and TJP3 KRT6A, KRT6B, KRT6C, KRT17, RHCG, SPRR1A, and VSNL1
Landscape of transcriptional deregulation in lung cancer (2018).	LAC SCC	TP63/SOX2/DMRT3 LEF1/MSC
Transcriptional regulatory networks in human lung adenocarcinoma (2012).	LAC	PPARG, CEBPB, ETV4, FLI1, TAL1, and NFκB1
Transcription Factor and lncRNA Regulatory Networks Identify Key Elements in Lung Adenocarcinoma (2018).	LAC	TP53, SMAD4, SOX9, NFE2L2, MGA, ETV6, GATA3, and RUNX1
Lineage transcription factors co-regulate subtype-specific genes providing a roadmap for systematic identification of small cell lung cancer vulnerabilities (2020).	SCLC	NKX2-1, PROX1, ASCL1, HES1, JAG2, TGFB2, FOXA2, ID2, INSM1, and PROX1.
Identification of Key Transcription Factors Associated with Lung Squamous Cell Carcinoma (2017).	LUSC	NFIC, BRCA1, NFATC2, IRF1, NR2F1, FOXF1, NR4A2, HOXA5, EGR1, EGR2, ZEB1, YY1, BRCA1, E2F3, and MEF2A.
Functional analysis of microRNA and transcription factor synergistic regulatory network based on identifying regulatory motifs in non-small cell lung cancer (2013).	NSCLC	E2F1, ESR1, STAT1, RB1, MYC, NFKB1, miR-590, and miR-570
TENET 2.0: Identification of key transcriptional regulators and enhancers in lung adenocarcinoma (2020).	LAC	NKX2-1, CENPA, FOXM1, and MYBL2
Systems-level network modeling of Small Cell Lung Cancer subtypes identifies master regulators and destabilizers (2019).	SCLC	ELF3 and NR0B1 as master regulators, and TCF3 as a master destabilizer. STAT6, and EBF1.

## Data Availability

All microarray datasets are fully available in Gene Expression Omnibus (GEO).

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
