# Peer review of "Identification of the Transcriptional Regulatory Role of RUNX2 by Network Analysis in Lung Cancer Cells"

_biomedicines, 2022, doi:10.3390/biomedicines10123122_

Round 1
Reviewer 1 Report
The manuscript entitled “Transcriptional regulatory role played by RUNX2 in lung cancer NSCLC” reports that RUNX2 has been identified as an important regulator of gene expression in non-small cell lung cancer (NSCLC) and may be an important biomarker for the development of specific therapies for lung diseases, including NSCLC. However, the following points must be addressed before the manuscript can be suitable for publication.
Comments:
1. The authors need to make specific changes to the title. For example: "Identification of the transcriptional regulatory role of RUNX2 by network analysis in lung cancer cells."
2. The authors need to indicate the criteria for dividing LC I and II in Table 1.
3. In Table 2, the authors also need to further analyze target genes of E2F3, FHL2, and TWIST1, which are coregulator with RUNX2.
4. In Table 3, the authors need to indicate and describe the signaling pathways in which the targets of RUNX2 are involved.
5. The authors need to indicate the meaning of red and blue spots in Figure 2.
Author Response
RESPONSES TO REVIEWERS: MANUSCRIPT ID: biomedicines-1986316
We want to thank the reviewers’ comments. Certainly, all helped to improve the document and clarify important issues. The responses were marked by the letter A and highlighted in red.
Reviewer 1
The manuscript entitled “Transcriptional regulatory role played by RUNX2 in lung cancer NSCLC” reports that RUNX2 has been identified as an important regulator of gene expression in non-small cell lung cancer (NSCLC) and may be an important biomarker for the development of specific therapies for lung diseases, including NSCLC. However, the following points must be addressed before the manuscript can be suitable for publication.
Comments:
- The authors need to make specific changes to the title. For example: "Identification of the transcriptional regulatory role of RUNX2 by network analysis in lung cancer cells."
A/ We changed the title as you properly suggested.
- The authors need to indicate the criteria for dividing LCI and II in Table 1.
A/ LCI is the coexpression network of dysregulated genes of lung cancer, which are not dysregulated in other lung diseases, while LCI is the coexpression network of dysregulated genes of lung cancer, which are not dysregulated in other types of cancer. The selection identified unique DEGs in lung cancer and helped to focus the analysis on lung cancer. We have clarified this important issue in section 3.2.
- In Table 2, the authors also need to further analyse target genes of E2F3, FHL2, and TWIST1, which are coregulator with RUNX2.
A/ We have added these three coregulators of RUNX2 in table 2. Therefore, we did the transcriptional regulatory network with the seven TFs.
- In Table 3, the authors need to indicate and describe the signalling pathways in which the targets of RUNX2 are involved.
A/ We have added signalling pathways related to every RUNX2 target gene according to DAVID and PANTHER analysis to table 3.
- The authors need to indicate the meaning of red and blue spots in Figure 2.
A/ We have added the meaning in figure 2 caption. In blue are the downregulated targets and in red are the upregulated targets.

Author Response
RESPONSES TO REVIEWERS: MANUSCRIPT ID: biomedicines-1986316
We want to thank the reviewers’ comments. Certainly, all helped to improve the document and clarify important issues. The responses were marked by the letter A and highlighted in red.
Reviewer 2
Comments to: "Transcriptional regulatory role played by RUNX2 in lung cancer NSCLC" the authors propose a research article regarding the putative involvement of RUNX2 transcription factor in lung cancer identified through a bioinformatic pipeline. From microarray dataset they selected a gene regulatory network of lung cancer with differentially expressed genes. From this resource transcription factors acting as main regulators and coregulators were selected and the binding motif on target gene identified. Among them RUNX2 is indicated as the most interesting regulator of the gene regulatory network selected. The article is an original research paper showing an interesting finding which need in the future to be experimentally validated for the development of any further research or clinical application. In addition, a more in-depth and comprehensive discussion of RUNX2 target genes would help to better delineate its role considering that it has been identified as a major regulator in cell of the osteoblastic lineage. In conclusion, the research plan and the article are well designed and realized, a better graphical display of the data in the tables would help to appreciate and improve the reported results.
A/ We have improved all tables information reported and clarified some important points.
COMMENTS Results,
paragraph 3.1 It is reported that 6 important transcriptional regulators were identified by NSCLC-GRN. Have you also identified these transcriptional factors among that analysed for the other lung pathologies in your previous work? Are these 6 factors exclusively identified by NSCLC-GRN? Have you identified common transcriptional regulators of NSCLC and other lung pathologies?
A/ In table 1, we show that E2F3 is coexpressed with BRCA1 in the common connectivity pattern between lung cancer datasets and other lung diseases, with FOXM1 in the common connectivity pattern between lung cancer datasets, RUNX1 in the coexpression networks of networks of unique lung cancer genes that are not deregulated in other lung diseases (LCI), or other types of cancer (LCII), and with RUNX2, FHL2 and TWIST1 in the gene regulatory network of NSCLC. We sum up the data for every transcription factor like this in table 1, highlighting the importance of RUNX2 since all our previous analysis. We add the coregulatory networks (New figure 2) in order to visualize the impact of the NSCLC-GRN regulators in the regulation of the transcription factors previously identifies in our coexpression studies. We can see now that the six regulators in our NSCLC-GRN are coregulators of previous transcription factors, therefore regulating transcription of lung cancer in cooperative and coordinated manner.
Paragraph 3.2 In my opinion the interesting results resumed in the table are not clearly exposed. First of all, I think it will be more useful to insert a title directly in the table. The titles in bold in the first line of the table are not homogeneous and generate confusion for the interpretation of the table. For example, the title of the first column is generically “TF” even though it refers to the 6 important transcription factors identified by NSLCGRN performed in this work. I think it should be written. On the contrary, the titles reported in the same line of “TF” all refer to other lung diseases. I think a way should be found to standardise the titles to make the reading of the table clearer. Furthermore, all the boxes show the names of transcription factors identified as coregulators, and expressed in other pathologies, of the 6 transcription factors identified by NSCLC-GRN. Instead, the acronyms in the last two boxes of the second column refer to pathologies. I think it would be better to indicate with a plus sign and write the acronym of the pathology with a different character than the one used for the transcription factors.
A/ In table 1, we have made important notation in the text and the table caption in order to make it understandable.
Paragraph 3.2 Line 248 Other than some transcription factors target of RUNX2 found in the RTN network, have you analyzed the others common genes? Have you classified these genes to understand whether these are involved in common pathways? And eventually which pathways? It might be useful to understand also the role of transcription factors identified by NSCLC-GRN. I think it would be useful to include a paragraph with the explanation on the biological role of genes regulated by RUNX2 other than transcriptional factors.
A/ We have included the signalling pathway analysis of all RUNX2 target genes identified in the transcriptional regulatory network.
Paragraph 3.3 In this paragraph potential target genes of RUNX2 were identified. All the target genes identified are transcriptional factors implicated in the regulation of lung disease and specifically in cancer? Have you identified target genes which are not transcription factors? As suggested before I think the table 3 should be improved to clearly read the results. I think it is important to highlight the biological role of the genes found as a function of the pathology in which they were identified. I do not think it is useful to report the name of the database that associated the targets gene with RUNX2 discussion In this section you discuss the role of RUNX2 transcription factor with its coregulators and cofactors. The biological role associated to this transcriptional factors is mainly related to the formation of the osteoblastic lineage but also with the regulation of lung cancer related genes. Have you analyzed whether common target genes are activated in the different pathway triggered by RUNX2 in combination with the different coregulators and cofactors?
A/ We have removed the column with the database, and we have included the signalling pathways related to every RUNX2 target, according to DAVID and PANTHER analysis to table 3. Moreover, we have included all the RUNX2 targets identified by the transcriptional regulatory network to the pathways analysis, in order to consider target genes that are not transcription factors, giving more support to the analysis related to the regulatory function of RUNX2 in lung cancer. We will consider in our future bioinformatic and experimental analysis if common target genes are activated in the different pathway triggered by RUNX2 in combination with the different coregulators and cofactors, we agree that it is a very important issue to take into account.

Reviewer 3 Report
The manuscript entitled: "Transcriptional regulatory role played by RUNX2 in the lung can- 2 cer NSCLC" investigated the good research problem. The work presented a very technical introduction and existing efforts in a good way. Methods and materials are fined and defined in a good way. However even though the manuscript is well written, but I have a few suggestions for further improvement in the current manuscript.
1. There should be more case studies in the work because it is hard for reader to understand too many technical things as defined in the manuscript.
2. All figures' visibility and resolutions are weak; it is good to use the best graphic tool and recall those images in pdf form.
3. The current work finding and limitations must be defined before the conclusion.
4. Add the comparing table of existing studies with the considered work in the same constraints as considered in the manuscript.
Author Response
RESPONSES TO REVIEWERS: MANUSCRIPT ID: biomedicines-1986316
We want to thank the reviewers’ comments. Certainly, all helped to improve the document and clarify important issues. The responses were marked by the letter A and highlighted in red.
Reviewer 3
The manuscript entitled: "Transcriptional regulatory role played by RUNX2 in the lung cancer NSCLC" investigated the good research problem. The work presented a very technical introduction and existing efforts in a good way. Methods and materials are fined and defined in a good way. However even though the manuscript is well written, but I have a few suggestions for further improvement in the current manuscript.
- There should be more case studies in the work because it is hard for reader to understand too many technical things as defined in the manuscript.
A/ we have included six case studies to analyse RUNX2 regulatory function in the discussion.
- All figures' visibility and resolutions are weak; it is good to use the best graphic tool and recall those images in pdf form.
A/ We have attached all the images in high resolution in a separate file for the journal.
- The current work finding, and limitations must be defined before the conclusion.
A/we have included the limitations found during the analysis in the conclusion section.
- Add the comparing table of existing studies with the considered work in the same constraints as considered in the manuscript.
A/There are two studies, one in LUAD and other in LUSC assessing the coexpression networks, we added to the beginning of the discussion. Moreover, there are eleven publications assessing the transcription regulatory network of lung cancer with different microarray and RNA-Seq studies, doing direct analysis and very different analysis on the dataset created or selected for the publication. None of them have tried to select deregulated genes unique to lung cancer, that are not deregulated in other lung diseases or other types of cancer, or does a coregulatory analysis, in order to study the cooperative and coordinated regulatory function of the transcription factors. However, some of the transcription factors identified by our bioinformatic pipelines also are important in some of the studies. We have made a table (New table 4) and added to the discussion, as you properly suggested.

Round 2
Reviewer 1 Report
The authors properly revised the manuscript.
Author Response
Thankyou very much for your observations,
Best regards
Adriana Rojas
Reviewer 3 Report
The current version of the manuscript is much improved as compared to first version. However, still few things need to be revised and addressed in the manuscript.
1. The time complexity of the manuscript is widely ignored in the manuscript
2. The finding and limitation of the proposed work is missing in the work.
3. There must be more case-studies related to the work.
4. The simulation environment is totally missing in the paper.
5. There should be comparison of the proposed work with the existing studies.
6. There should be numeric comparison of the studies.
Author Response
RESPONSES TO REVIEWER: MANUSCRIPT ID: biomedicines-1986316
We want to thank the reviewer comments. Certainly, all helped to improve the discussion and conclusion, clarifying the complex timeline of the paper. The responses were marked by the letter A and highlighted in red.
The current version of the manuscript is much improved as compared to first version. However, still few things need to be revised and addressed in the manuscript.
- The time complexity of the manuscript is widely ignored in the manuscript
A/ We have added this important point at the beginning of the discussion, stablishing a timeline of the bioinformatic pipeline (Lines 334-381).
- The finding and limitation of the proposed work is missing in the work.
A/ We have added the main findings considering the limitations of the study, at the of the end of the discussion and conclusion (Lines 556-583).
- There must be more case-studies related to the work.
A/ We have added more case-studies related to the gene regulatory function and coregulatory complex formation of RUNX2 (Lines 493-501, 518-531).
- The simulation environment is totally missing in the paper.
A/ We have already published the simulation environment in reference 26, and the TRN and CoRegNet code is published in for every library in Bioconductor.
- There should be comparison of the proposed work with the existing studies.
A/ We have added the comparison of our work with previous studies in the discussion, based on the main goal of the methodology used to assess the heterogeneity of lung cancer (Lines 383-397).
- There should be numeric comparison of the studies.
A/ We have done a numeric comparison of the studies in the discussion (Lines 383-397).
